# Vector-Based Magnetic Circuit Modelling of Induction Motors

**Braden Kidd**

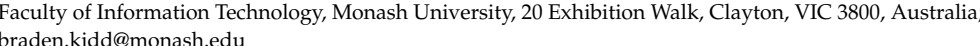

Faculty of Information Technology, Monash University, 20 Exhibition Walk, Clayton, VIC 3800, Australia; braden.kidd@monash.edu

**Abstract:** Electro-mechanical devices incorporating rotating magnetic fields can be modelled using a wide range of analytical techniques. Choosing a modelling technique usually requires a trade off between computational efficiency and accuracy. Magnetic flux-based models aim to achieve an optimum balance between computational intensity and accuracy, as required for real time control applications. This paper demonstrates how vector-based magnetic circuit equations can be used to describe the operational characteristics of an induction motor at a more fundamental level than commonly used magnetic flux models. Doing so allows for closed form equations to be derived directly from device-specific geometry. The resultant model has advantages of numerical method-based analytical techniques while retaining the computational efficiency of closed form equations.

**Keywords:** magnetic circuit; rotating magnetic fields; direct quadrate model; magnetic equivalent circuits; induction motor

## 1. Introduction

Modelling electromagnetic devices can be achieved using simple analytical techniques such as those based on electrical and magnetic equivalent circuits. The elementary circuit elements upon which these equivalent circuits are based were originally derived to describe the characteristics of electromagnetic devices with no relative constituent motion. As such, using them to model devices incorporating rotational motion often requires a high degree of abstraction.

In the case of electrical equivalent circuits such as the Steinmetz equivalent circuit of an induction motor, parameters can either be empirically derived [1,2], or derived using more complex modelling techniques [3]. This results in a black box model, where internal workings and parameters are unavailable. The models' computational efficiency and the ability to extrapolate its usage to non-ideal operating conditions [4] has kept it relevant despite the availability of alternatives. However, the lack of access to internal dynamic parameters limits its ability to be extrapolated to describe more complex operational behaviour in its basic form [5].

A more fundamental approach to modelling electrical machines can be achieved using numerical method-based computer simulations at the expense of greater computational complexity. These modelling techniques divide the internal geometry of the simulated device into discrete elements, each with their own unique electrical, magnetic and mechanical properties. Common examples of this approach are variations of the Lumped Parameter Model (LPM) [6]. These models use an interconnected mesh of magnetic circuit elements to calculate magnetic flux. In its most basic application, only linear magnetic field behaviour can be modelled. However, its application can be extended to include modelling non-linear behaviour such as magnetic saturation [7], fault detection [8] and thermal analysis [9].

The Finite Element Model (FEM) is another commonly used numerical method technique. It works by solving partial differential equations derived directly from Maxwell's equations at the boundaries of each discrete element. This is a more fundamental approach to modelling electrical and magnetic fields compared to LPM and is therefore regarded as

being more accurate [10,11]. A trade off for this improved accuracy is increased computation time [12]. Hybrid FEM models also exist that reduce computational complexity using either equivalent circuits [13] or lumped parameters [14].

A compromise between computationally intensive numerical methods and equivalent circuits can be achieved using magnetic flux-based models such as those based on the Direct Quadrate (D-Q) model. These models define internal parameters such as flux producing currents and torque producing currents [15]. By modelling internal dynamic factors, a more detailed operational model can be implemented relative to high-level equivalent circuits [16,17], with less computational complexity than numerical method-based techniques.

In this paper, a set of partial differential equations describing a rotating magnetic field and magnetic circuit power transfers will be defined. These equations describe the relationship between magnetic flux and its time derivative. Power transfers due to the magnetic flux-based variables and reluctance are also described. Using these equations and the analytical technique presented in this paper, it is possible to calculate inductive power transfers between rotating objects. These equations can be applied to the analysis of any AC machine incorporating a rotating magnetic field, including variable reluctance configurations.

To demonstrate the validity of this analytical technique, these magnetic circuit equations will be applied to the steady state analysis of an induction motor with results benchmarked against those derived using FEM. It will be based on a more detailed electric motor model than that used to derive the D-Q model. This approach results in a less abstract model that can be derived in terms of closed form equations. While magnetic flux model parameters are usually derived based on dynamic simulations [18,19], the proposed analytical technique parameters will be derived based on motor geometry and the electrical and magnetic material properties.

The proposed analytical technique will only be applied to a specific case in this paper involving steady state analysis. This will, however, provide the basis upon which further applications of the underlying equations can be used to model more complex operational modes, such as those involving transients and faults.

## 2. Elementary Definitions

To derive a magnetic circuit model of an induction motor, it is necessary to define key parameters to be measured and to describe their behaviour in a environment experiencing circular motion. These techniques are analogous to using circular motion equations to describe rotating objects. Although Newton's second law of motion can be directly applied to a rotating object, using equations of circular motion can considerably reduce the analytical complexity of describing such an object. Similarly, using circular motion magnetic circuit equations can simplify the analysis of rotating electromagnetic devices to the measurement of a few key parameters.

There are four key parameters to be measured when analysing an electromagnetic device incorporating circular motion. Electric current and voltage are two electrical domain parameters with magnetic flux and the time derivative of magnetic flux being their corresponding magnetic domain parameters. In this analysis, electrical domain parameters are mapped onto magnetic domain parameters, then, the analysis is performed in the magnetic domain. To define the relationship between the two magnetic domain parameters, consider the vector representation of a rotating magnetic field as described by (1). This magnetic flux vector is defined using a $i, j, k$ coordinate system, with a radian frequency of $\omega$, a phase offset of $\theta$ and time represented by the variable $t$.

$$\mathbf{\Phi} = |\mathbf{\Phi}| \cos(\omega t + \theta)\hat{i} + |\mathbf{\Phi}| \sin(\omega t + \theta)\hat{j} \tag{1}$$

Differentiating (1), with respect to time, allows the derivative of the magnetic flux to be expressed in vector form. For the general case, both the magnitude $|\mathbf{\Phi}|$ and the phase $\theta$ will be considered implicit functions of time and the radian frequency $\omega$ will be a constant. Using these definitions, the time derivative of (1) can be calculated to be (2).

$$\frac{\partial \mathbf{\Phi}}{\partial t} = \left(\frac{d|\mathbf{\Phi}|}{dt}\cos(\omega t + \theta) - |\mathbf{\Phi}|(\omega + \frac{d\theta}{dt})\sin(\omega t + \theta)\right)\hat{i}$$
$$+ \left(\frac{d|\mathbf{\Phi}|}{dt}\sin(\omega t + \theta) + |\mathbf{\Phi}|(\omega + \frac{d\theta}{dt})\cos(\omega t + \theta)\right)\hat{j} \tag{2}$$

Taking the cross and dot product of the magnetic flux and its time derivative allows (3) and (4), respectively, to be derived.

$$\mathbf{\Phi} \times \frac{\partial \mathbf{\Phi}}{\partial t} = |\mathbf{\Phi}|^2(\omega + \frac{d\theta}{dt}) \tag{3}$$

$$\mathbf{\Phi} \cdot \frac{\partial \mathbf{\Phi}}{\partial t} = |\mathbf{\Phi}|\frac{d|\mathbf{\Phi}|}{dt} \tag{4}$$

It is also possible to use the fundamental properties of the dot and cross product to derive an equation that is independent of the angle between the magnetic flux and its time derivative. This involves squaring both sides of (3) and (4), then adding them together, resulting in (5).

$$\left|\frac{\partial \mathbf{\Phi}}{\partial t}\right|^2 = \left(\frac{d|\mathbf{\Phi}|}{dt}\right)^2 + |\mathbf{\Phi}|^2(\omega + \frac{d\theta}{dt})^2 \tag{5}$$

Another definition required for this analysis is an angular dependant measurement of resistance. To define this quantity, consider the two conductors coloured grey in Figure 1.

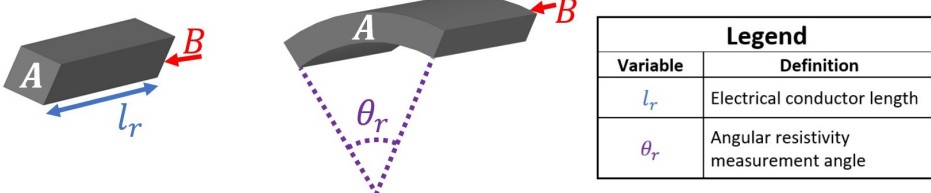

| Legend | |
|---|---|
| **Variable** | **Definition** |
| $l_r$ | Electrical conductor length |
| $\theta_r$ | Angular resistivity measurement angle |

**Figure 1.** Two electrical conductors with dimensions.

In Figure 1, the electrical current flows through each conductor coloured in grey between the surfaces labelled *A* and *B*. Assuming the conductive material is homogeneous throughout each volume, the resistance of the left most conductor can be calculated by multiplying its resistance per unit length $R_l$ by its length $l_r$. Defining the resistance per unit length provides a simple technique to calculate the conductor resistance for an arbitrary length. A similar process can be applied to a change in angular dimension, as demonstrated in the right most conductor. Defining a parameter $R_\theta$ with units of Ohm-radians allows the resistance of the right most conductor $R$ between the surfaces labelled *A* and *B* to be calculated using (6), where $\theta_r$ is defined in Figure 1.

$$R = \frac{R_\theta}{\theta_r} \tag{6}$$

Even if the conductor properties are not homogeneous, $R_l$ can still be used if the conductor is made of a series connection of identical segments, provided the resistance is measured for an integer number of those segments. Similarly, (6) is still valid in situations where periodic angular repetitions occur, provided the resistance is measured for integer multiples of each identical angular segment.

During the operation of an induction motor, energy is transferred from the stator to the rotor via electromagnetic induction. Quantifying the energy transfers due to this process can be achieved using vector-based magnetic circuit equations. The power transfer equation for magnetic circuits [20] is shown in (7).

$$P = \Re\left(\mathbf{\Phi} \cdot \frac{\partial \mathbf{\Phi}}{\partial t}\right) + \frac{1}{2}\frac{d\Re}{dt}\left(\mathbf{\Phi} \cdot \mathbf{\Phi}\right) \tag{7}$$

For the analysis of an induction motor, magnetic reluctance will be assumed as constant, thereby resulting in the second product term of (7) to equal zero. Therefore, magnetic circuit power transfers can be calculated based on reluctance and both the magnitude and angle of magnetic flux and its time derivative vectors.

Observing magnetic flux vectors from different rotating frames of reference will result in (7) evaluating different values. It has been demonstrated that the induction motor rotor torque can be calculated using the difference in power transfer to the rotor obtained by evaluating (7) from the rotor's and stator's frame of reference [20].

Therefore, frame dependant quantities will be identified in this paper using a right vertical line with a subscript identifying its frame of reference. For example, $\mathbf{\Phi}|_{stator}$ and $\mathbf{\Phi}|_{rotor}$ are magnetic flux vectors evaluated from the stator's and rotor's frame, respectively. While the magnitude of magnetic flux vectors is frame independent in this analysis, the magnitude and vector value of the magnetic flux time derivative is frame dependant.

For the induction motor derived in this analysis, all rotor electric current-induced magnetic flux will be represented as the single vector $\mathbf{\Phi}_r$ and all stator current-induced magnetic flux will be represented by the vector $\mathbf{\Phi}_s$. These two vectors are added together using vector addition to form the net magnetic flux $\mathbf{\Phi}_n$. Rotor power transfers will then be calculated by substituting the rotor magnetic flux vector into the magnetic flux term in (7) and the time derivative of the net magnetic flux into the flux partial differential term in (7). Repeating this process using stator magnetic flux instead of rotor magnetic flux is used to calculate stator power transfers. A single and constant value of reluctance derived by analysing the magnetic field energy in the motor will be used. Evaluating these power transfers from different rotating frames can then be used to derive an operational model of the induction motor.

## 3. Rotor and Stator Magnetic Flux Calculations

In an induction motor, magnetic flux is produced by electric currents flowing in the stator windings and inducing currents in the rotor bars. To calculate the rotor's magnetic flux due to induced currents, consider two opposite rotor bars from a cage rotor, as shown in Figure 2.

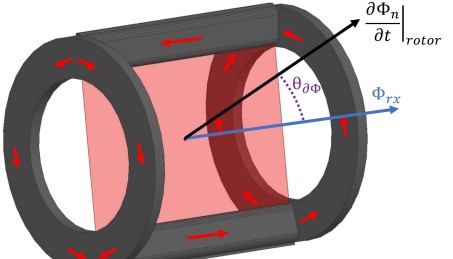

| Legend | |
|---|---|
| **Variable** | **Definition** |
| $\left.\dfrac{\partial \Phi_n}{\partial t}\right|_{rotor}$ | Net Magnetic Flux time derivative evaluated from the rotor frame |
| $\theta_{\partial \Phi}$ | Angular displacement from the partial derivative of the net magnetic flux vector |
| $\Phi_{rx}$ | Dual rotor bar magnetic flux |

**Figure 2.** Two opposite rotor bars with superimposed vectors.

In Figure 2, the electric currents' direction is shown by arrows on the rotor conductors and $\mathbf{\Phi}_{rx}$ is the resultant magnetic flux. This magnetic flux will flow through the shaded rectangle with the time derivative of the net magnetic flux from the rotor's frame of reference determining the induced voltage around the conduction loop. This induced voltage $V_{rx}$ can be calculated to be (8), where $\theta_{\partial \Phi}$ is the angular displacement from the partial derivative of the net magnetic flux vector.

$$V_{rx} = \left|\frac{\partial \mathbf{\Phi}_n}{\partial t}\right|_{rotor} \cos(\theta_{\partial \Phi}) \tag{8}$$

The induced current in rotor bar pairs can be determined by dividing the induced voltage $V_{rx}$ by the resistance around the conduction loop. The conduction loop resistance, and by extension the rotor's magnetic flux, will depend on the number of rotor bars. When the number of rotor bars is high, this analysis can be simplified using a homogenised model

of the rotor bars with the rotor resistance measured in Ohm-radians. To illustrate how this is achieved, consider the cross section of rotor bars as shown in the left diagram in Figure 3.

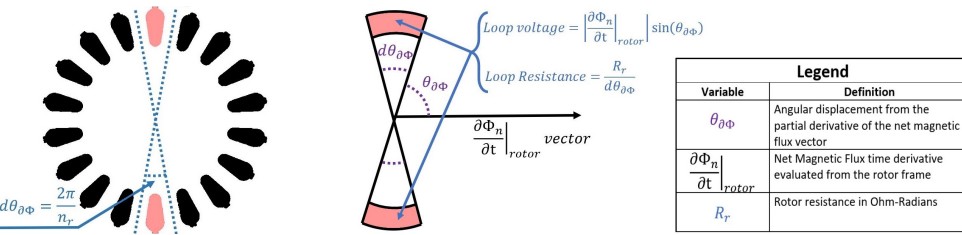

**Figure 3.** Rotor bar cross sections.

The angular displacement between rotor bars is $\frac{2\pi}{n_r}$, where $n_r$ is the number of rotor bars. As each rotor bar has identical dimensions and angular displacement from its proceeding rotor bar, it is possible to define an angular resistance in Ohm-radians for the rotor $R_r$ by dividing the resistance of the current path as shown in Figure 2 by the angular displacement between rotor bars. This allows the resistance of a conductive loop for a small differential change in angular displacement $d\varphi_\Phi$ to be described, as also shown in Figure 3.

Dividing the Figure 3 loop voltage by loop resistance allows the current bounded by the angular displacement of $d\theta_{\partial\Phi}$, as represented by the variable $i_{seg}$, to be calculated.

$$i_{seg} = \left|\frac{\partial\boldsymbol{\Phi_n}}{\partial t}\right|_{rotor} \frac{\sin(\theta_{\partial\Phi})}{R_r} d\theta_{\partial\Phi} \tag{9}$$

The total rotor magnetic flux can be determined by combining the contribution of each rotor current segment from Figure 3 into a single magnetic flux vector $\boldsymbol{\Phi_r}$. This cannot be achieved using a vector addition, as the magnetic field from each rotor segment impacts the other segments. To account for this effect, consider the magnetic field due to two differential pairs of rotor bars and their resultant field directions, as shown in Figure 4.

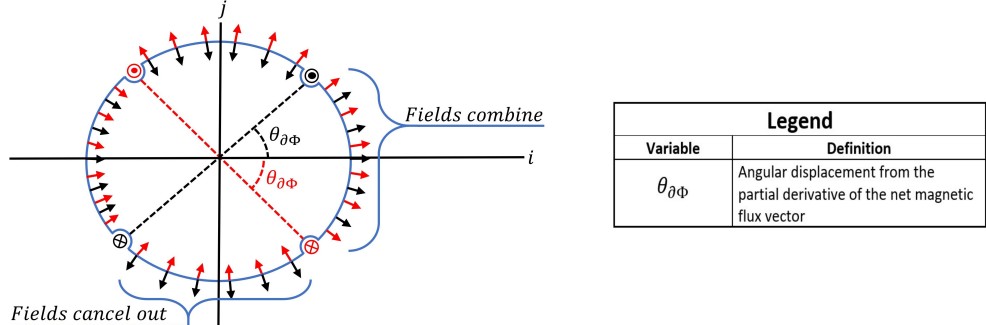

**Figure 4.** Two differential currents with resultant field directions.

The fields from each current pair will spread approximately evenly around the air gap between the rotor and stator to follow the course of least reluctance. When two current pairs are at the same angular displacement from the *i* axis in Figure 4, components of their fields cancel out while some field components combine. Based on this principle and the geometry from Figure 4, the magnetic flux due to two current pairs $\boldsymbol{\Phi_{rxd}}$ can be calculated to be (10), where $i_{seg}$ is the current magnitude flowing in each current pair.

$$|\boldsymbol{\Phi_{rxd}}| = \frac{4i_{seg}\theta_{\partial\Phi}}{\pi\Re} \tag{10}$$

Assuming the *i* axis from Figure 4 is aligned with the derivative of the net magnetic flux vector, Equation (10) can be used to calculate the magnetic flux flowing through the rotor using the (9) value of current magnitude.

$$|\mathbf{\Phi}_{rxd}| = \left|\frac{\partial\mathbf{\Phi}_n}{\partial t}\right|_{rotor} \left|\frac{4\theta_{\partial\Phi}\sin(\theta_{\partial\Phi})}{\pi R_r \Re}d\theta_{\partial\Phi}\right| \tag{11}$$

Integrating (11) over the range of $0 \le \theta_{\partial\Phi} \le \frac{\pi}{2}$ as shown in Figure 4, results in the (12) value of rotor magnetic flux.

$$|\mathbf{\Phi}_r| = \frac{4}{\pi R_r \Re}\left|\frac{\partial\mathbf{\Phi}_n}{\partial t}\right|_{rotor}\right| \tag{12}$$

The $\mathbf{\Phi}_r$ magnetic flux described by (12) was calculated in the direction of the $\left.\frac{\partial\mathbf{\Phi}_n}{\partial t}\right|_{rotor}$ vector, as shown in the right picture in Figure 3. This is true by definition, as the *i* axis in Figure 4 is aligned with this vector. It can also be observed in Figure 4 that for any value of $\theta_{\partial\Phi}$, the net magnetic flux along the *j* axis will be zero. This will result in no component of the rotor magnetic flux vector being perpendicular to the time derivative of the net magnetic flux vector as evaluated in the rotor frame.

As (12) was calculated using the rotor resistance in Ohm-radians, it is independent of the specific number of rotor bars. However, the stator winding configuration cannot be homogenised. Therefore, the relationship between stator currents and induced magnetic flux must be derived separately for every stator winding configuration. For the purpose of demonstrating the model in this paper, a two pole, three phase induction motor stator with 24 rotor slots will be analysed.

Each stator slot contains windings from two phases. For sinusoidal three phase current flowing through the stator windings, the total current amplitude flowing through each stator slot $i_s$ can be calculated to be (13), where $i_p$ is the phase current amplitude and $n_s$ is the number of turns per phase per stator slot.

$$i_s = \sqrt{3}i_p n_s \tag{13}$$

There are 12 different phase slot winding configurations arranged in groups of two. The difference in the current phase between each adjacent group of two is 30°. Based on this information, the (10) formula for magnetic flux can also be applied to stator currents. Substituting (13) into (10) for each differential stator slot pair allows (14) to be derived.

$$|\mathbf{\Phi}_s| = 8.3301\frac{i_p n_s}{\Re} \tag{14}$$

As the number of stator turns per slot per phase $n_s$ and magnetic reluctance $\Re$ are constant in this application, (14) can be simplified using the variable $K_i$ for further analysis, as defined in (15).

$$|\mathbf{\Phi}_s| = K_i i_p \tag{15}$$

## 4. Modelling Magnetic Reluctance

The conversion of the current into magnetic flux requires the magnetic path reluctance to be known. The magnetic reluctance is calculated using magnetic field energy within the motor as a function of the net magnetic flux $\mathbf{\Phi}_n$. Equation (16) will be used for this calculation, where *U* is the magnetic field energy.

$$\Re = \frac{2U}{|\mathbf{\Phi}_n|^2} \tag{16}$$

The motor will be divided into 5 regions to calculate magnetic field energy. These concentric regions are shown in the Figure 5 motor cross section, cut perpendicular to the axis of rotation.

In Figure 5, the inner rotor region that falls within the radius of $r_i$ is the section of the rotor that does not contain the rotor bars. This will be modelled with a constant magnetic permeability and parallel magnetic field lines. The total net magnetic flux $\mathbf{\Phi_n}$ flows through this region. Due to the assumed parallel nature of the magnetic field lines, the magnetic flux density will be modelled as constant throughout this region. The magnetic flux density within this region can be calculated to be (17), assuming that all of the net magnetic flux $\mathbf{\Phi_n}$ passes through this region and the rotor has an axial length of $l_m$.

$$B_{inner\_rotor} = \frac{|\mathbf{\Phi_n}|}{2r_i l_m} \tag{17}$$

Integrating the inner rotor region energy due to the (17) magnetic flux density and substituting this value into (16) allows the (18) value of reluctance to be calculated.

$$\mathfrak{R}_{inner\_rotor} = \frac{\pi}{4\mu_0 \mu_r l_m} \tag{18}$$

The rotor bar region and the stator slot region contain both electrical and magnetic field conductors. To make this model easily transferable between different motors, these regions will be modelled homogeneously with each differential angular segment containing an electrical and magnetic conductive region. To achieve this, the ratio of the rotor bar region cross sectional area that contains the rotor bars, relative to its total area, will be defined as $C_r$.

$$C_r = \frac{rotor\ bar\ area}{rotor\ bar\ region\ area} \tag{19}$$

To illustrate how this concept can be used to calculate the magnetic field energy within the rotor bar region of the rotor, consider the small angular segment of this region bounded by the angular displacement of $d\theta_\Phi$, where $\theta_\Phi$ is the angular displacement from the net magnetic flux vector.

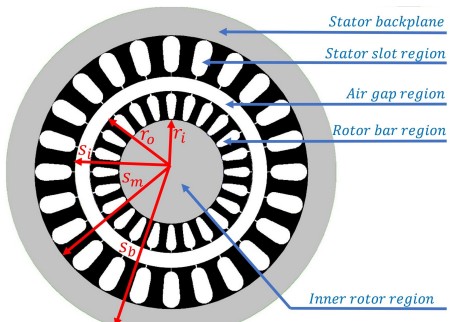

| Legend | |
|---|---|
| **Variable** | **Definition** |
| $r_i$ | Inner rotor radius |
| $r_o$ | Outer rotor radius |
| $s_i$ | Stator inner radius |
| $s_m$ | Stator middle radius |
| $s_b$ | Stator backplane radius |

**Figure 5.** Motor cross section.

In Figure 6, the top shaded region represents the electrical conductor and the lower shaded region represents the magnetic field conductor within this angular segment. Magnetic flux will pass through both the electrical and magnetic conductive regions, although most magnetic flux will pass through the magnetic conductor. The ratio of magnetic flux passing through the electrical to magnetic flux region can be calculated using magnetic circuit equations based on the parallel connection of reluctance elements. Based on this model, the magnetic flux density within the magnetic conductive region $B_{mcr}$ can be calculated as a function of distance from the rotational axis $r$ to be (20).

$$B_{mcr} = \frac{|\mathbf{\Phi_n}|\mu_r \cos(\theta_\Phi)}{2l_m r(C_r + \mu_r(1 - C_r))} \tag{20}$$

Integrating this energy over the rotor slot area results in the (21) value of energy within the rotor bar region's magnetic conductors $U_{mcr}$.

$$U_{mcr} = \frac{\pi \mu_r |\mathbf{\Phi_n}|^2 (1 - C_r)}{8 \mu_0 l_m (C_r + \mu_r (1 - C_r))^2} log_e \left( \frac{r_o}{r_i} \right) \tag{21}$$

The magnetic flux density in the electrical conductive region $B_{ecr}$ of the rotor bar region can be calculated using the same process to be (22).

$$B_{ecr} = \frac{|\mathbf{\Phi_n}| \cos(\theta_\Phi)}{2 l_m r (C_r + \mu_r (1 - C_r))} \tag{22}$$

Integrating this energy over the rotor slot area results in the (23) value of energy within the rotor bar region's electric conductors $U_{ecr}$.

$$U_{ecr} = \frac{\pi |\mathbf{\Phi_n}|^2 C_r}{8 \mu_0 l_m (C_r + \mu_r (1 - C_r))^2} log_e \left( \frac{r_o}{r_i} \right) \tag{23}$$

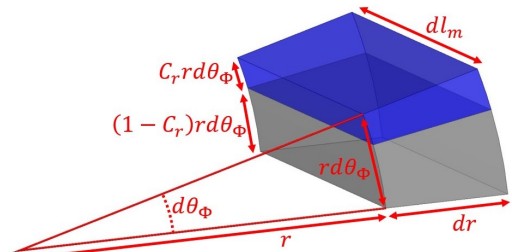

| Legend | |
|---|---|
| **Variable** | **Definition** |
| $r$ | Radius variable used for integration |
| $\theta_\Phi$ | Angular displacement from net magnetic flux |
| $l_m$ | Length of rotor and stator |
| $C_r$ | Rotor bar area to total area ratio |

**Figure 6.** Rotor bar region bound by small angular displacement.

The sum of (21) and (23) is the total magnetic field energy within the rotor bar region of the rotor. Substituting this sum into (16) allows the reluctance of the rotor bar region $\Re_{rotor\_bar}$ to be calculated.

$$\Re_{rotor\_bar} = \frac{\pi}{4 \mu_0 l_m (C_r + \mu_r (1 - C_r))} log_e \left( \frac{r_o}{r_i} \right) \tag{24}$$

Calculating the air gap reluctance can be achieved using this same process. Alternatively, substituting $C_r = 1$ into (24) is valid for the air gap region due to its lack of magnetically permeable material. The radius variables $r_o$ and $r_i$ from (24) need to be changed to $s_i$ and $r_o$, respectively, as per the air gap dimensions in Figure 5, resulting in the (25) value of air gap reluctance.

$$\Re_{air\_gap} = \frac{\pi}{4 \mu_0 l_m} log_e \left( \frac{s_i}{r_o} \right) \tag{25}$$

The stator slot region reluctance is calculated using the same technique used to derive (24). This requires the cross sectional area of the stator slots divided by the cross sectional area of the entire stator slot region $C_s$ to be defined.

$$C_s = \frac{stator\ slot\ area}{stator\ slot\ region\ area} \tag{26}$$

Repeating the rotor bar region reluctance calculation using the stator slot area dimensions results in the (27) value of reluctance.

$$\Re_{stator\_slot} = \frac{\pi}{4 \mu_0 l_m (C_s + \mu_r (1 - C_s))} log_e \left( \frac{s_m}{s_i} \right) \tag{27}$$

Magnetic flux in the back plane will be the sum of all the magnetic flux that has entered minus what has left. This can be calculated as a function of angular displacement from the net magnetic flux vector $\theta_\Phi$ by integrating the contribution of each differential flux component leaving the stator slot region.

$$\Phi_{s\_backplane} = \int_{\theta_\Phi}^{0} \frac{|\mathbf{\Phi_n}|dl_m}{2l_m} \cos(\theta_\Phi)d\theta_\Phi \tag{28}$$

$$\Phi_{s\_backplane} = \frac{|\mathbf{\Phi_n}|}{2l_m} \sin(\theta_\Phi)dl_m \tag{29}$$

Dividing (29) by the stator back plane area allows the magnetic flux density in this region to be calculated.

$$B_{s\_backplane} = \frac{|\mathbf{\Phi_n}|}{2l_m(s_o - s_m)} \sin(\theta_\Phi) \tag{30}$$

Integrating the magnetic field energy due to (30) over the entire stator back plane region and substituting the resultant energy value into (16) results in the (31) value of stator back plane magnetic reluctance.

$$\Re_{s\_backplane} = \frac{\pi(s_o + s_m)}{8\mu_0\mu_r(s_o - s_m)l_m} \tag{31}$$

The total equivalent magnetic reluctance $\Re$ of the motor can be calculated by summing together all reluctance elements as described by (32).

$$\Re = \Re_{inner\_rotor} + \Re_{rotor\_bar} + \Re_{air\_gap} + \Re_{stator\_slot} + \Re_{s\_backplane} \tag{32}$$

## 5. Combining Magnetic and Electric Circuit Power Transfers

Magnetic flux from the rotor and stator will add together to form the net magnetic flux. Due to the sinusoidal distribution of magnetic flux as a function of angular displacement, the stator and rotor magnetic flux vectors can be added using a vector addition.

In a steady state operation of the induction motor, it can be assumed that the net magnetic flux magnitude and the relative angle with its time derivative vector are constant. Therefore, the time derivative of the phase variable $\theta$ from (5) can be set to zero. When the derivative of the magnetic flux magnitude is zero, Equation (4) states the angle between the magnetic flux, and its time derivative is 90°. In this situation, Equation (5) can be simplified to equal (33).

$$\left|\frac{\partial\mathbf{\Phi_n}}{\partial t}\right| = |\mathbf{\Phi_n}|\omega \tag{33}$$

The 90° angle between the net magnetic flux and its time derivative will be the same for all rotating frames. However, the net magnetic flux derivative's magnitude is frame dependant due to the $\omega$ term in (33).

It was demonstrated in Section 3 that the rotor's induced magnetic flux is parallel to the derivative of the net magnetic flux vector. Using this information, the angular relationship between the stator, rotor and net magnetic flux vectors and the derivative of the net magnetic flux can be depicted in vector form, as shown in Figure 7.

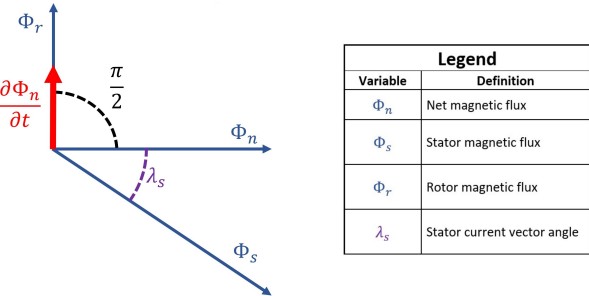

**Figure 7.** Induction motor magnetic flux vectors.

The stator magnetic flux vector $\mathbf{\Phi_s}$ and the rotor magnetic flux vector $\mathbf{\Phi_r}$ add together to form the net magnetic flux vector $\mathbf{\Phi_n}$. As these vectors represent fields with a sinusoidal air gap distribution relative to angular displacement, they can be summed together using a vector addition. Using the vector geometry from Figure 7, it is possible to derive (34).

$$|\mathbf{\Phi_n}| = |\mathbf{\Phi_s}| \cos(\lambda_s) \tag{34}$$

The magnitude of the net magnetic flux can also be expressed in terms of the phase current amplitude $i_p$ using (15) and (34).

$$|\mathbf{\Phi_n}| = K_i i_p \cos(\lambda_s) \tag{35}$$

Induced voltage is another quantity from the electrical domain that has a magnetic domain equivalent. Just as the variable $K_i$ can be used to calculate stator current from the stator magnetic flux, the variable $K_v$ can be used to calculate the derivative of the net magnetic flux from induced voltage. If the stator winding voltage was only due to the rotating magnetic field, Equation (36) would describe this relationship, where $V_s$ is the stator phase voltage amplitude.

$$\left| \frac{\partial \mathbf{\Phi_n}}{\partial t}^* \right| = K_v V_s \tag{36}$$

However, Equation (36) cannot be exact, as induced voltages exist in the stator windings due to the resistive losses and inductance due to stray magnetic fields. The stator winding voltage due to resistive losses can be calculated by multiplying the stator winding resistance per phase $R_s$ by the stator phase current $i_s$. As this product is a voltage, it can be converted into the magnetic domain by multiplying by $K_v$ to equal $K_v i_s R_s$. This vector quantity will have a 180° phase shift relative to the stator magnetic flux vector $\mathbf{\Phi_s}$ as per electrical circuit laws. The induced voltage due to the derivative of the net magnetic flux vector, as shown in Figure 7 and described by (33), will have a 180° phase shift as per Lenz's law. Figure 8 shows this information on a vector diagram.

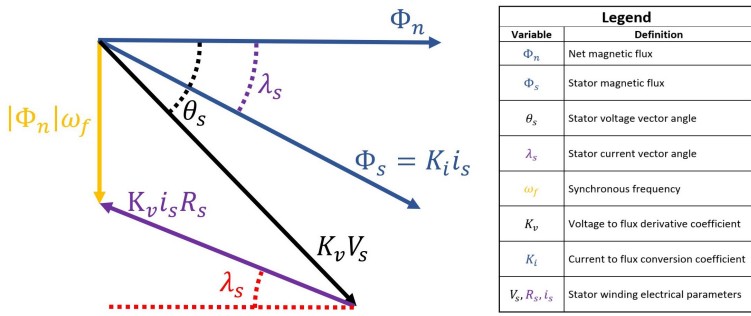

**Figure 8.** Induction motor magnetic flux and voltage vectors.

In Figure 8, the angle $\theta_s$ is the angle between the net magnetic flux and the magnetic flux derivative vector based only on the externally applied stator winding voltage. As this

vector diagram is viewed from the stator's frame of reference, the net winding voltage when resistive induced voltages are subtracted from the applied voltage is a function of $\omega_f$, as shown by the left most vector in Figure 8.

Some of the stator windings magnetic flux does not pass through the rotor and therefore cannot be included as part of the stator magnetic flux vector $\boldsymbol{\Phi_s}$. These stray fields can be modelled as inductors in the series with the stator windings, with an effective inductance of $L_s$. This effect requires the stator winding voltage drop due to both the resistive losses and stray field inductance $V_{R+L}$ to be accounted for, as described by (37).

$$V_{R+L} = i_s \sqrt{R_s^2 + (\omega_f L_s)^2} \tag{37}$$

The induced winding voltage, as described by (37), will also have a phase shift relative to the purely resistive current induced voltage, as depicted in Figure 8. This additional phase shift is represented by the variable $\rho_s$, as defined by (38).

$$\rho_s = \tan^{-1}\left(\frac{\omega_f L_s}{R_s}\right) \tag{38}$$

Incorporating this additional information into Figure 8 results in Figure 9.

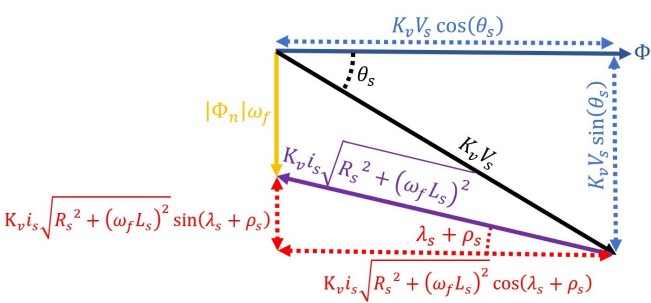

**Figure 9.** Induction motor voltage vectors.

Equations (39) and (40) can be derived from the Figure 9 vector geometry.

$$V_s \cos(\theta_s) = i_s \sqrt{R_s^2 + (\omega_f L_s)^2} \cos(\lambda_s + \rho_s) \tag{39}$$

$$V_s \sin(\theta_s) = i_s \sqrt{R_s^2 + (\omega_f L_s)^2} \sin(\lambda_s + \rho_s) + \frac{|\boldsymbol{\Phi_n}|\omega_f}{K_v} \tag{40}$$

The angle between the $K_v V_s$ and $K_i i_s$ vectors from Figure 8 is the same as the phase angle between the AC voltage and current. This is because transferring quantities between the electrical and magnetic domain does not change phases in this example. The AC power delivered to the motor from a three phase source $P_{supply}$ can be calculated to be (41) from elementary electrical identities.

$$P_{supply} = \frac{3}{2} V_s i_s \cos(\theta_s - \lambda_s) \tag{41}$$

Substituting Equations (39) and (40) into (41) results in the (42) value of electrical power.

$$P_{supply} = \frac{3}{2} R_s i_s^2 + \frac{3\omega_f}{2K_v} |\boldsymbol{\Phi_n}| i_s \sin(\lambda_s) \tag{42}$$

Equation (35) can be substituted into (42) to eliminate $i_s$ in the second product term, resulting in the (43) value of $P_{supply}$.

$$P_{supply} = \frac{3}{2} R_s i_s^2 + \frac{3\omega_f}{2 K_v K_i} |\mathbf{\Phi_n}|^2 \tan(\lambda_s)$$ (43)

The product of the variables $K_i$ and $K_v$ can be determined using the transforming properties of electrical to magnetic domain conversions. Equation (41) is the power transferred to the stator in the electrical domain. Equating this to its equivalent magnetic domain power transfer equation using (15) and (36) allows the product term $K_i K_v$ to be calculated.

$$\frac{3}{2} V_s i_s \cos(\theta_s - \lambda_s) = \Re \mathbf{\Phi_s} \cdot \frac{\partial \mathbf{\Phi_n}}{\partial t}^*$$ (44)

$$K_i K_v = \frac{3}{2\Re}$$ (45)

Substituting (45) into (43) results in the (46) value of stator electrical power.

$$P_{supply} = \frac{3}{2} R_s i_s^2 + \Re \omega_f |\mathbf{\Phi_n}|^2 \tan(\lambda_s)$$ (46)

## 6. Power Losses

Power supplied to the induction motor can be described by (46), although at this stage in the derivation, variables $\lambda_s$, $i_s$ and $\mathbf{\Phi_n}$ are unknown. To calculate these variables, all sources of internal losses need to be quantified. Magnetic losses resulting from eddy currents and hysteresis losses can be quantified using the Steinmetz's equation. For this model, stator magnetic power losses $P_{sm}$ will be approximated using (47), while rotor magnetic losses $P_{rm}$ will be approximated using (48).

$$P_{sm} = M_{se} \omega_f^2 |\mathbf{\Phi_n}|^2 + M_{sh} \omega_f |\mathbf{\Phi_n}|^2$$ (47)

$$P_{rm} = M_{re} \omega_s^2 |\mathbf{\Phi_n}|^2 + M_{rh} \omega_s |\mathbf{\Phi_n}|^2$$ (48)

In Equations (47) and (48), the variables $M_{se}$ and $M_{re}$ are the coefficients of stator eddy current losses and rotor eddy current losses respectively. $M_{sh}$ and $M_{rh}$ are the coefficients of stator hysteresis losses and rotor hysteresis losses, respectively.

Rotor electrical power losses can be calculated using the magnetic circuit power transfer equation, as described by (7). Substituting the (12) value of the rotor magnetic flux and the (33) derivative of the magnetic flux evaluated from the rotor's reference frame into (7) results in the (49) value of the rotor electric power loss $P_{re}$.

$$P_{re}|_{rotor} = \frac{4}{\pi R_r} \omega_s^2 |\mathbf{\Phi_n}|^2$$ (49)

Evaluating $P_{re}$ from the stator's reference frame results in the (50) value of power loss.

$$P_{re}|_{stator} = \frac{4}{\pi R_r} \omega_s \omega_f |\mathbf{\Phi_n}|^2$$ (50)

The difference between the rotor electric power loss when evaluated from the stator's and rotor's frame of reference is the rate electrical energy is converted into mechanical energy due to the rotor torque. Only electrical losses need to be accounted for in this calculation, as they produce the mutually coupled magnetic flux between the rotor and stator.

Based on this information, the power transfer due to torque on the rotor can be calculated to be (51), where $\omega_r$ is the rotor rotational speed ($\omega_f - \omega_s$).

$$P_\tau = \frac{4}{\pi R_r} \omega_s \omega_r |\mathbf{\Phi_n}|^2$$ (51)

## 7. Complete Power Transfer Equations

Equation (46) quantifies the rate electrical energy is transferred to the motor and is made up of two product terms. The first product term describes energy lost due to the electrical resistance of the stator windings, while the second describes inductive energy transfers. These inductive energy transfers are the (50) rotor power and the (47) and (48) magnetic losses. Equating the second product term of (46) to equal these inductive energy transfers allows the value of $\tan(\lambda_s)$ to be calculated.

$$\tan(\lambda_s) = \frac{4\omega_s}{\pi R_r \Re} + \frac{M_{se}\omega_f^2 + M_{sh}\omega_f + M_{re}\omega_s^2 + M_{rh}\omega_s}{\omega_f \Re} \tag{52}$$

The value of $\lambda_s$ can now be calculated using (52), as it is a function of known constants and externally measurable variables.

It is also necessary to calculate the magnitude of the net magnetic flux $\boldsymbol{\Phi_n}$. This can be achieved by first squaring (39) and (40), then adding them together to eliminate their dependence on $\theta_s$.

$$V_s^2 = i_s^2\left(R_s^2 + (\omega_f L_s)^2\right) + \left(\frac{\omega_f|\boldsymbol{\Phi_n}|}{K_v}\right)^2 + \frac{2i_s\omega_f}{K_v}\sin(\lambda_s + \rho_s)\sqrt{R_s^2 + (\omega_f L_s)^2} \tag{53}$$

Substituting Equations (35), (38) and (45) into Equation (53) allows the $i_s$, $\rho_s$ and $K_v$ terms to be eliminated, resulting in (54).

$$\frac{V_s^2}{|\boldsymbol{\Phi_n}|^2} = \frac{R_s^2 + (\omega_f L_s)^2}{K_i^2}\left(\tan^2(\lambda_s) + 1\right) + \left(\frac{2\Re K_i\omega_f}{3}\right)^2 + \frac{4\Re\omega_f}{3}\left(R_s\tan(\lambda_s) + \omega_f L_s\right) \tag{54}$$

As the amplitude of the stator winding voltage $V_s$ is known, Equation (54) can be used to calculate $|\boldsymbol{\Phi_n}|$. This value can then be substituted into (51) to calculate the power transfer due to the rotor torque. Electrical power supplied to the motor can be calculated using (46) and the stator phase current using (35).

## 8. FEM Simulation Parameters

The equations derived in this paper can be used to predict the torque speed characteristics of an induction motor. To validate the model derived in this paper, a FEM simulation of an induction motor's torque speed characteristics will be used to establish a benchmark against which the accuracy of these equations can be assessed. The following geometric parameters as described in Section 4 of this paper used in the FEM simulation are as follows.

- $r_i$ = 43 mm
- $r_o$ = 65 mm
- $s_i$ = 65.5 mm
- $s_m$ = 87 mm
- $s_b$ = 112 mm
- $l_m$ = 160 mm
- $c_r$ = 0.65
- $c_s$ = 0.65
- $\mu_r$ = 3000

The relative permeability of non-oriented superior motor grade electrical steel is used in the simulation due to its popularity in small- to mid-sized motors [21]. Soft magnetic materials such as silicon steel (3.2 wt% Si) are increasing in popularity due to low loss factors. However, high processing costs and the difficulty in optimising all required mechanical and electromagnetic parameters is still a limiting factor in the commercialisation of soft magnetic materials [22]. As this analysis does not require a complex magnetic model to

demonstrate its validity, the magnetic properties of conventional motor grade electrical steel will be used where required.

Inserting these dimensions into the Section 4 magnetic reluctance equations results in the following reluctance values.

- $\Re_{inner\_rotor}$ $= 1302 \text{ H}^{-1}$
- $\Re_{rotor\_bar}$ $= 1536 \text{ H}^{-1}$
- $\Re_{air\_gap}$ $= 29{,}933 \text{ H}^{-1}$
- $\Re_{stator\_slot}$ $= 1055 \text{ H}^{-1}$
- $\Re_{s\_backplane}$ $= 5182 \text{ H}^{-1}$

The sum of these motor reluctance values results in $\Re = 39{,}008 \text{ H}^{-1}$.

Not all the magnetic flux produced by stator currents is mutually coupled with the rotor. This results in stator leakage flux. Quantifying this value is required to calculate the stator windings leakage inductance. Leakage flux can be quantified using FEM [23] or real time emulation models [24]. Magnetic circuit-based leakage flux models are also effective and can be derived based on geometric factors [25]. As such, the leakage inductance will be calculated in this model using a reluctance mesh.

This value is determined by estimating the magnetic path reluctance encountered by these stray fields. The path encountered by one set of stray fields is shown in Figure 10 by the reluctance circuit surrounding a stator slot.

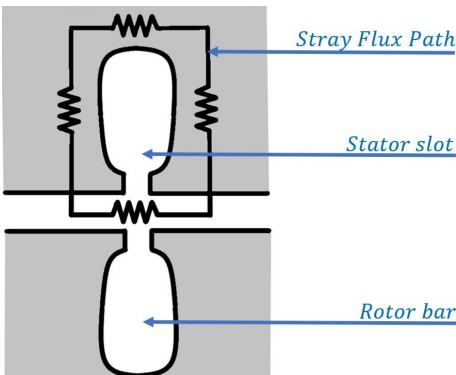

**Figure 10.** Stray fields' magnetic flux path.

Figure 10 depicts the reluctance circuit surrounding a single stator slot. As a phase winding current will flow through four adjacent slots, the parallel connection of four reluctance circuits as depicted by Figure 10 will need to be calculated. Using the resulting reluctance mesh, the reluctance encountered by stray flux is estimated to be $2.85 \times 10^6 \text{ H}^{-1}$. The induction motor is wound with 26 windings per phase per stator slot. This results in 104 conductors enclosed by the stray fields' reluctance mesh. Therefore, the effective series inductance in the stator windings due to leakage flux can be calculated using (55).

$$L = \frac{n^2}{\Re} \tag{55}$$

In (55), $L$ is inductance, $n$ is the number of turns and $\Re$ is reluctance. Using this relationship, the inductance due to a single phase winding passing through four adjacent stator slots is 3.80 mH. As each phase has four sets of these windings, the leakage inductance per stator phase winding, as represented by the variable $L_s$, is 15.2 mH.

The rotor resistance can be calculated by measuring the electrical path resistance of a current loop encompassing two opposite rotor bars, as shown in Figure 2. This resistance for the simulated rotor is estimated to be $1.14 \times 10^{-4} \ \Omega$. Multiplying this value by the angular displacement between rotor bars $\left(\frac{\pi}{12}\right)$ results in the rotor resistance $R_r$ value of $2.98 \times 10^{-5}$ Ohm-radians. The stator winding resistance $R_s$ has been set in the simulation

to be 1.616 Ω. Other simulation parameters are a phase to phase RMS voltage of 400 V, a Wye connection and a synchronous frequency of $100\pi$ radians per second.

For this simulation, all magnetic loss coefficients will be set to zero. This is to validate the model derivation up to this point. Deriving magnetic loss coefficients directly from motor geometry and magnetic material properties will require an analysis of approximately the same length as this paper up to this point. Alternatively, using FEM to derive approximations for magnetic loss parameters would be using FEM to validate its own modelling technique. This approach would also invalidate the claim in the introduction that all parameters can be derived directly from physical laws without the aid of existing simulation models.

## 9. FEM Simulation Results

Predictions made by the magnetic circuit-based model presented in this article will be benchmarked against a commercially available FEM model to assess its accuracy. This is achieved using the FEM simulation tools of Ansys Maxwell 16.0. A 3D rendering of the simulated motor model is shown in Figure 11 from two angles with phase windings colour-coded black, red and blue for phases 1, 2 and 3, respectively. The motor model is cut both radially and axially, as the entire motor geometry is not required to be simulated due to symmetries.

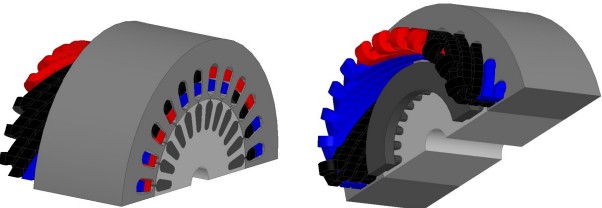

**Figure 11.** Three-dimensional rendering of FEM-simulated induction motor.

Using the parameters detailed in the previous section, a FEM simulation of the induction motor was undertaken to obtain its output power, phase current and efficiency as a function of rotor rotational speed. A 2D rendering of the FEM simulated motor's magnetic fields is shown in Figure 12 once they have reached their steady state values.

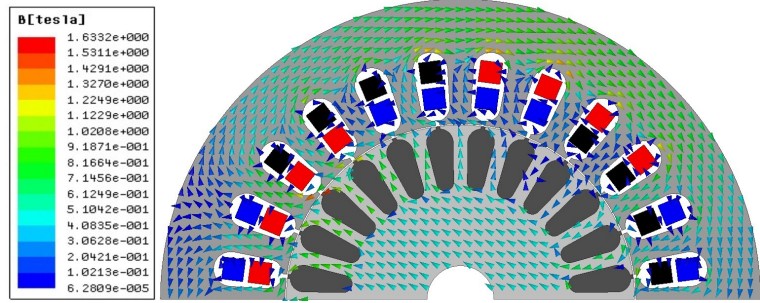

**Figure 12.** Two-dimensional rendering of FEM-simulated induction motor with superimposed fields.

This rendering validates the assumptions of the magnetic field distribution within the motor, as stated in Section 4. Magnetic field distribution within the inner rotor region is approximately uniform in both magnitude and direction. Stator fields also vary in intensity, reaching their peak values perpendicular to the direction of the inner rotor's magnetic field from the axis of rotation.

The FEM simulation also allowed the output power as a function of rotational speed to be calculated, as shown in Figure 13. These simulated values are compared to those obtained using the magnetic circuit model derived in this paper, with the rotational speed measured in Revolutions Per Minute (RPM).

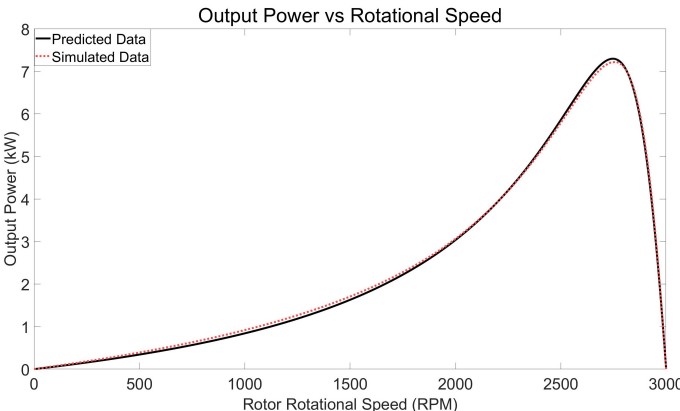

**Figure 13.** Output power vs. rotor rotational speed.

The simulated output power in Figure 13 slightly exceeds the predicted value for rotational speeds between 500 RPM and 2000 RPM. However, the maximum predicted output power was 7.29 kW, which is 70 W more than the simulated value of 7.22 kW. For rotational speeds above 2800 RPM, which correspond to typical operating conditions, the FEM-simulated and magnetic circuit predictions are closely matched. The RMS deviation between the two output power data sets is 59.3 W or 2.47% of the average predicted value.

As the magnetic circuit parameters derived in this model do not correspond directly with those used in the FEM simulation, it is difficult to identify which parameter caused this discrepancy. It is worth noting that although the FEM-simulated magnetic losses were set to zero in the material settings, a low but non zero value of magnetic loss was recorded. This implies that the FEM model uses magnetic loss correction techniques to improve its accuracy in practical settings.

The predicted phase current in Figure 14 is less than the FEM-simulated values for rotational speeds below 1500 RPM. Both data sets are closely matched for rotational speeds above 1500 RPM with only a slight divergence around 2400 RPM. Overall, the RMS deviation between the two data sets is 0.223 A or 0.87% of the average predicted current.

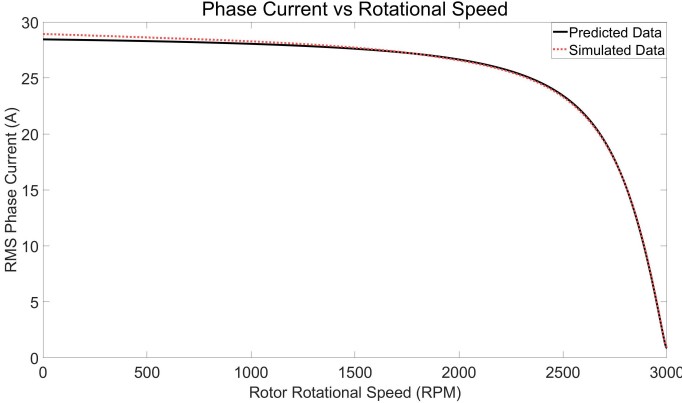

**Figure 14.** Phase current vs. rotor rotational speed.

The predicted motor efficiency in Figure 15 was slightly less than its simulated value for most rotational speeds. However, the peak predicted efficiency of 95.8% exceeds the maximum simulated efficiency of 94.3%. Overall, the RMS deviation between the two data sets is 0.92%.

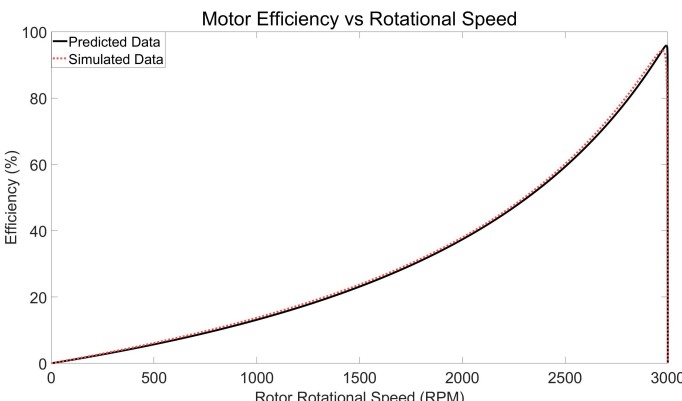

**Figure 15.** Motor efficiency vs. rotor rotational speed.

Figure 15 also reveals another possible reason for the differences between the FEM and magnetic circuit-derived results. The FEM efficiency is lower than magnetic circuit efficiency close to the synchronous frequency. In this operational region, the stator losses dominate; thus, FEM stator losses might be higher than the magnetic circuit model. As overall FEM efficiency is slightly higher, a larger FEM inductive coupling factor or lower FEM rotor losses may also be responsible.

The efficiency function in Figure 15 increases with rotational speed across most of the data set range, before quickly reducing to zero at the synchronous speed. This overall shape is to be expected and can be inferred from the rotor power transfer Equations (50) and (51). Only factoring these into the efficiency equation allows an upper limit on motor efficiency to be derived. This upper limit is (56), where $\eta$ is the motor efficiency.

$$\eta \leq 1 - \frac{\omega_s}{\omega_f} \tag{56}$$

In (56), $\omega_s$ is the slip frequency, which is the synchronous frequency minus the rotational frequency. Equation (56) represents the theoretical limit of the inductive transfer efficiency to the rotor with a constant reluctance. To achieve this efficiency, no stator losses can be incurred. As such, the deviation in efficiency from (56) can reveal the impact of stator related and mechanical sources of power loss on overall motor efficiency.

It is also possible to calculate the angle between the net and stator magnetic flux vectors $\lambda_s$ as a function of rotor rotational speed. As this is not a variable commonly measured by simulation software, plotting this information offers a unique insight into the internal magnetic field dynamics within the induction motor. The angle $\lambda_s$ is an important parameter, as it describes how much stator magnetic flux, and therefore, the stator winding current, is needed to produce the net magnetic flux, as per (34). It also determines how much stator winding current is required to achieve inductive power transfers. This value can be determined by substituting (34) into (46).

The value of $\lambda_s$ is shown in Figure 16 as a function of rotor rotational velocity.

It can be observed from Figure 16 that the angle between the stator and net magnetic flux vectors $\lambda_s$ is close to 90° for low rotor speeds and quickly approaches 0° near the synchronous rotational speed. It is also interesting to note that the operating speeds most likely to be encountered under normal load have the most variation in $\lambda_s$.

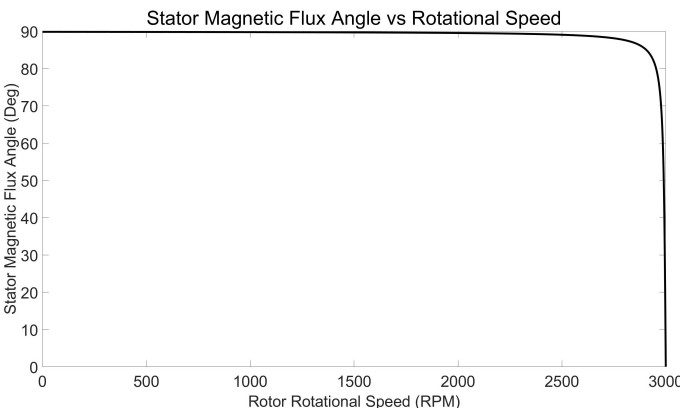

**Figure 16.** Angle between net and stator magnetic flux vectors vs. rotor rotational speed.

Figure 16 also reveals that the relative angles between magnetic flux vectors can change with rotational speed. Other magnetic flux models such as the D-Q model have a fixed angle of 90° between their two magnetic flux vectors. As the angle between the rotor and net magnetic flux vectors is 90°, it is possible to see why this assumption might still result in accurate observations. However, as per (4), this constant 90° angle is only possible when the derivative of the net magnetic flux magnitude is zero, as the rotor magnetic flux is parallel to the time derivative of the net magnetic flux. Any change to net magnetic flux magnitude will also change this angle, irrespective of the angular frequency $\omega$ and phase $\theta$, as (4) is independent of these variables. Accounting for this change in angle between magnetic flux vectors could therefore provide this model with an advantage over fixed angle models.

The net magnetic flux magnitude $|\mathbf{\Phi}_n|$ can be readily obtained from the magnetic circuit equations derived in this paper. This value is shown in Figure 17 as a function of rotor rotational speed.

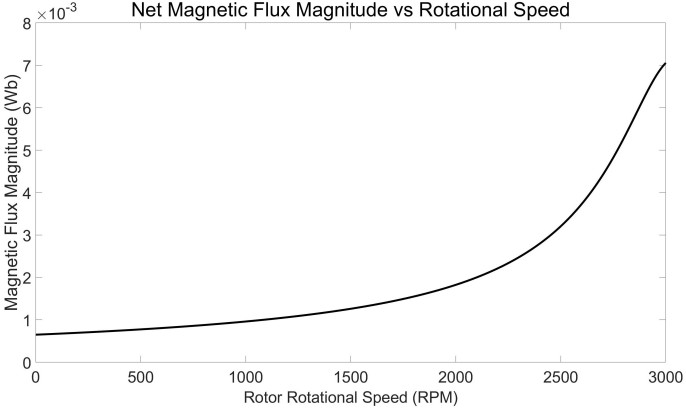

**Figure 17.** Net magnetic flux magnitude vs. rotor rotational speed.

Figure 17 depicts a continual increase in the net magnetic flux magnitude as a function of rotor rotational speed. The stator magnetic flux, which is directly proportional to the stator current, as shown in Figure 14, only decreases when the rotational speed increases. This illustrates the impact the angle between the net and stator magnetic flux has on the operation of the induction motor. For low speeds, the stator and rotor magnetic flux are almost opposite each other and therefore much of their magnetic flux is canceled out. As the angle $\lambda_s$ decreases, less stator flux and therefore phase current is required to produce the net magnetic flux.

## 10. Conclusions

This paper has demonstrated how magnetic circuit analysis can be applied to situations involving rotating magnetic fields using energy and circular motion-based equations. The key components of this analytical technique involves both magnetic flux and the time derivative of magnetic flux in vector form. Quantifying these magnetic domain quantities from their electrical domain equivalents and using the mathematical relationship between them allows for the derivation of closed form equations. The resultant magnetic circuit model can then be used to describe the motor's operational characteristics and provide insights into parameters not usually quantified in traditional models.

As a magnetic flux-based model, the proposed analytical technique bears some similarities to the D-Q model. One advantage of this model over the D-Q model is that parameters can be derived directly from motor geometry. Another is that it is derived at a lower level of abstraction than the assumptions upon which the D-Q model is based. As such, it involves a more fundamental approach, from the application of the laws of physics to the description of AC machines.

It has also been demonstrated in this paper that magnetic circuit analysis can be used to describe rotating machines without the need for ad hoc or unexplained variables to match predictions with observations. The same parameters and variables were used to derive all of the predicted data without the need for individual correction coefficients for each measurements.

Due to the closed form equations derived using this analytical technique, there is an inherent computational speed advantage of this model over numerical method-based models. This could be advantageous for future applications of the underlying mathematical relationships described in this paper if implemented in real time control applications. It can also provide motor designers with information such as relative magnetic flux vector angles, which cannot be readily obtained from some design software.

The model presented in this paper only represents one specific application of these equations relating the magnetic flux and its time derivative. Therefore, there are still further applications of this research using the underlying equations and analytical techniques to derive dynamic models incorporating more complex losses. There is also the potential to use these magnetic circuit equations as a basis on which to build control algorithms, which has not yet been explored. Through further applications of the equations derived in this paper, it may be possible to use magnetic circuit analysis to achieve a more optimal balance between computational efficiency and accuracy in practical applications.

**Funding:** This research was funded through the provision of an Australian Government Research Training Program Scholarship, a Monash University Faculty of Information Technology Research Scholarship and a CSIRO Data61 Scholarship.

**Conflicts of Interest:** The author declares no conflict of interest. The funders had no role in the design of the study; in the collection, analyses, or interpretation of data; in the writing of the manuscript, or in the decision to publish the results.

## Abbreviations

| Symbol | Quantity |
| --- | --- |
| $B$ | Magnetic flux density |
| $B_{inner\_rotor}$ | Inner rotor magnetic flux density |
| $B_{s\_backplane}$ | Stator backplane magnetic flux density |
| $B_{ecr}$ | B field in rotor bar area electric conductor |
| $B_{mcr}$ | B field in rotor bar area magnetic conductor |
| $C_r$ | Rotor bar area to total area ratio |
| $C_s$ | Stator slot area to total area ratio |

| | |
|---|---|
| $i_p$ | Phase current amplitude |
| $i_s$ | Stator slot current amplitude |
| $i_{seg}$ | Current in differential rotor segment |
| $K_i$ | Current to flux conversion coefficient |
| $K_v$ | Voltage to flux derivative coefficient |
| $L$ | Inductance value |
| $l_m$ | Length of rotor and stator |
| $l_r$ | Electrical conductor length |
| $L_s$ | Stator winding inductance |
| $M_{re}$ | Coefficient of rotor eddy current losses |
| $M_{rh}$ | Coefficient of rotor hysteresis losses |
| $M_{se}$ | Coefficient of stator eddy current losses |
| $M_{sh}$ | Coefficient of stator hysteresis losses |
| $n$ | Number of winding turns |
| $n_s$ | Winding turns per stator slot per phase |
| $P$ | Generic power variable |
| $P_{re}$ | Rotor electrical power loss |
| $P_{rm}$ | Rotor magnetic eddy current losses |
| $P_{rotor}$ | Power transfer to the rotor |
| $P_{sm}$ | Stator magnetic eddy current losses |
| $P_{stator}$ | Power transfer from stator |
| $P_{supply}$ | Electrical power supplied to motor |
| $P_\tau$ | Mechanical power transfer due to rotor torque |
| $r$ | Radius variable used for integration |
| $r_i$ | Inner rotor radius |
| $r_o$ | Outer rotor radius |
| $R$ | Generic resistance value |
| $R_l$ | Linear resistivity |
| $R_r$ | Rotor angular resistance |
| $R_s$ | Stator winding resistance |
| $R_\theta$ | Angular resistivity |
| $\Re$ | Magnetic reluctance |
| $\Re_{air\_gap}$ | Air gap reluctance |
| $\Re_{inner\_rotor}$ | Inner rotor reluctance |
| $\Re_{rotor\_bar}$ | Rotor bar area reluctance |
| $\Re_{s\_backplane}$ | Stator backplane reluctance |
| $\Re_{stator\_slot}$ | Stator slot reluctance |
| $S_b$ | Stator backplane radius |
| $S_i$ | Stator inner radius |
| $S_m$ | Stator middle radius |
| $t$ | Time |
| $U$ | Generic energy variable |
| $U_{ecr}$ | Energy in rotor bar area electric conductor |
| $U_{mcr}$ | Energy in rotor bar area magnetic conductor |
| $V_{rx}$ | Induced voltage in rotor bar pair x |
| $V_{R+L}$ | Stator resistive and inductive voltage |
| $V_s$ | Stator voltage amplitude |
| $\eta$ | Rotor power transfer efficiency |
| $\theta$ | Generic angle variable |
| $\theta_r$ | Angular resistivity measurement angle |
| $\theta_{\partial\Phi}$ | Angular displacement from $\mathbf{\Phi}_n$ partial derivative |
| $\theta_\Phi$ | Angular displacement from $\mathbf{\Phi}_n$ |
| $\theta_s$ | Stator voltage vector angle |
| $\lambda_s$ | Stator current vector angle |
| $\mu_r$ | Relative permeability |
| $\mu_0$ | Vacuum permeability |
| $\rho_s$ | Inductive stator voltage phase shift |

| $\Phi$ | Generic magnetic flux variable |
|---|---|
| $\Phi_n$ | Net magnetic flux |
| $\Phi_s$ | Stator magnetic flux |
| $\Phi_r$ | Rotor magnetic flux |
| $\Phi_{rx}$ | Dual rotor bar magnetic flux |
| $\Phi_{rxd}$ | Differential rotor magnetic flux |
| $\Phi_{s\_backplane}$ | Stator backplane magnetic flux |
| $\omega$ | Generic frequency variable |
| $\omega_f$ | Synchronous frequency |
| $\omega_s$ | Slip frequency |
| $\omega_r$ | Rotor rotational frequency |

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
