# Peer review of "Vector-Based Magnetic Circuit Modelling of Induction Motors"

_2673-8724, doi:10.3390/magnetism2020010_

Round 1
Reviewer 1 Report
The theoretical part of the article is well professionally and methodically processed. However, the article has relatively few references (20) and all are used only on the first three pages of the text.
There are a few minor formal errors in the text, such as the numbering of equations (16) and (46) - lines 208 and 357 are given without parentheses.
In my opinion, however, the use of FEM is insufficiently described in the article. In which software was the FEM method used?
The article lacks a better illustration of the magnetic flux distribution in the machine according to FIG. 5 at least for its basic modes, for example, such as the QuickField SW allows. I mean something like in the attached picture for the FEM solution of a synchronous machine.
The permeability value of µr 4000 (data on page 14) is, in my opinion, very high, because the value of about 1000 is often used when respecting shape demagnetization.
Reviewer 2 Report
Dear Authors,
thank you for the presentation of the vector based magnetic circuit modelling of induction motors.
Please add legends in Figures: 1, 2, 3, 4, 5, 6, 7, 8, 9, 10.
The Figure 10 is not clear. What do you want to present? Add more information in the Figure 10.
Round 2
Reviewer 2 Report
Review Report
18th of March, 2022
Summary: This paper demonstrates how vector based magnetic circuit equations can be used to describe the operational characteristics of an induction motor at a more fundamental level than commonly used magnetic flux models. The key components of this analytical technique involves both magnetic flux and the time derivative of magnetic flux in vector form. Quantifying these magnetic domain quantities from their electrical domain equivalents and using the mathematical relationship between them, allows for the derivation of closed form equations.
Advantage: The resultant model has advantages of numerical method based analytical techniques while retaining the computational efficiency of closed form equations.
Conclusion: Your paper was improved. Thank you for the presentation of the vector based magnetic circuit modelling of induction motors.